# ALIGNED BETTER, LISTEN BETTER FOR AUDIO-VISUAL LARGE LANGUAGE MODELS

**Yuxin Guo**[1,2,3]**, Shuailei Ma**[4,5]**, Shijie Ma**[1,2]**, Xiaoyi Bao**[1,2,3]**. Chenwei Xie**[3]**,
Kecheng Zheng**[4]**, Tingyu Weng**[3]**, Siyang Sun**[3]**, Yun Zheng**[3]**, Wei Zou**[1,2*]
[1]School of Artificial Intelligence, University of Chinese Academy of Sciences
[2]MAIS, Institute of Automation, Chinese Academy of Sciences (CASIA)
[3]Tongyi Lab, Alibaba Group    [4]Ant Group    [5] Northeastern University
{guoyuxin2021, wei.zou}@ia.ac.cn

## ABSTRACT

Audio is essential for multimodal video understanding. On the one hand, video inherently contains audio, which supplies complementary information to vision. Besides, video large language models (Video-LLMs) can encounter many audio-centric settings. However, existing Video-LLMs and Audio-Visual Large Language Models (AV-LLMs) exhibit deficiencies in exploiting audio information, leading to weak understanding and hallucinations. To solve the issues, we delve into the model architecture and dataset. (1) From the architectural perspective, we propose a fine-grained AV-LLM, namely Dolphin. The concurrent alignment of audio and visual modalities in both temporal and spatial dimensions ensures a comprehensive and accurate understanding of videos. Specifically, we devise an audio-visual multi-scale adapter for multi-scale information aggregation, which achieves spatial alignment. For temporal alignment, we propose audio-visual interleaved merging. (2) From the dataset perspective, we curate an audio-visual caption & instruction-tuning dataset, called AVU. It comprises 5.2 million diverse, open-ended data tuples (video, audio, question, answer) and introduces a novel data partitioning strategy. Extensive experiments show our model not only achieves remarkable performance in audio-visual understanding, but also mitigates potential hallucinations.

## 1 INTRODUCTION

Humans perceive the dynamic world through their eyes and ears, with visual and auditory information complementing each other, both are indispensable. Similarly, the audio modality proves crucial for the comprehensive understanding capabilities of Multimodal Large Language Models (MLLMs) (Liu et al., 2023; Ma et al., 2024c; Bao et al., 2024; Wu et al., 2025; Ma et al., 2025a). On the one hand, audio modality can provide complementary information to the visual modality (Guo et al., 2024b), aiding MLLMs in more accurate comprehension. On the other hand, there are many audio-centric tasks in audio-visual (AV) understanding, *e.g.*, AV question answering, AV source localization, AV event localization, and AV segmentation (Tian et al., 2018; Yang et al., 2022; Guo et al., 2023; Mo & Morgado, 2022; Guo et al., 2024a). However, most existing Video-LLMs directly neglect audio, with only a small portion incorporating both visual and audio modalities. A natural question arises: *How proficient are these models in their audio-visual comprehension capabilities?*

To answer the question, we design two progressive experiments, as in Figure 1. First, we evaluate the AV understanding abilities of current AV-LLMs, *e.g.*, Video-LLaMA (Zhang et al., 2023a) and Video-LLaMA 2 (Cheng et al., 2024). We found they consistently neglect audio and the descriptions solely come from the visual content. Furthermore, when directly querying the audio content, the responses are always the speculations and associations derived from the visual input, rather than the audio itself. For example, when Video-LLaMA is presented with a cooking video, the response is "background music playing throughout the video." But actually, the audio features a male commentator's narration. Besides, when replacing the background sound with white noise, the model's responses remain unchanged, indicating that the model does not extract information from the audio.

---

*Corresponding author.

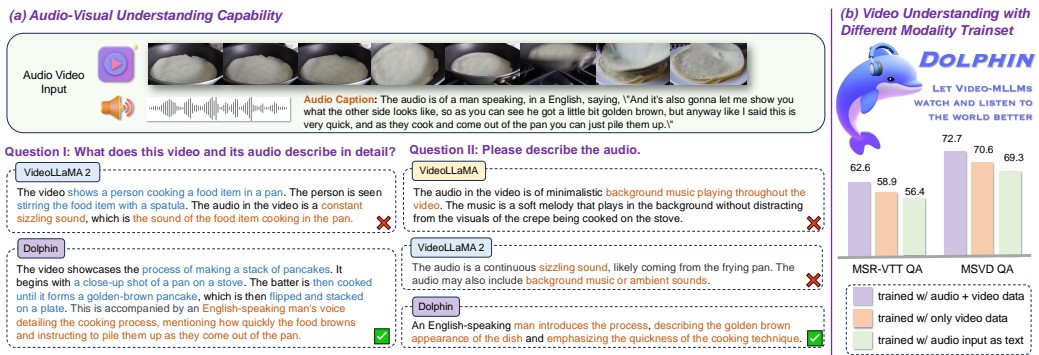

Figure 1: (a) Audio visual capability of previous AV-LLMs and our Dolphin. We pose questions separately for audio-video and audio, discovering that VideoLLaMA and VideoLLaMA 2 exhibit significant hallucinations for audio understanding, while Dolphin produces accurate responses. (b) Audio could provide complementary information compared to video. Incorporating audio into training greatly enhances video understanding.

Based on the results, it naturally begs the question: *Why do AV-LLMs tend to neglect the audio modality?* We attribute the following reasons: **(1) For alignment**, most models lack fine-grained alignment and interactions between modalities, and simply concat the visual and audio tokens. **(2) For datasets size**, large-scale audio-visual instruction-following datasets are scarce, and most works align vision-language and audio-language separately, resulting in less coordinated audio-video representations, **(3) In current datasets, visual modality has relatively higher information density**, where audio does not provide necessary content, consequently, AV-LLMs tend to disregard audio.

To solve these problems, we explore from two perspectives, *i.e.*, model architecture and training dataset. From the model perspective, we propose a novel fine-grained AV-LLM, namely Dolphin, which aligns audio and visual modalities both spatially and temporally and effectively harnesses both complementary modalities. For spatial alignment, we propose an audio-visual multi-scale adapter, which extracts multi-scale features and implements audio-visual interaction and merging at various scales. For temporal alignment, we propose audio-visual interleaved merging, where both audio and visual serve as context for each other through interleaved tokens. Finally, the fine-grained tokens aligned both spatially and temporally are projected into the input space of LLM to achieve remarkable joint audio-visual understanding.

From the dataset perspective, we propose a large-scale audio-visual understanding caption and instruction-following dataset, called AVU, which consists of 5.2M AV captions and question-and-answer (Q&A) tuples. We extract video and audio meta-information and generate high-quality captions and Q&A pairs. The dataset is divided into several splits according to AV consistency for different training objectives. Specifically, we incorporate datasets from several AV tasks, *e.g.*, AVE (Tian et al., 2018), AVL (Mo & Morgado, 2022), AVS (Zhou et al., 2022) and AVVP (Tian et al., 2020), and convert them to fine-grained instruction-following data. Besides, some negative samples for rejection are devised to avoid potential hallucinations. To comprehensively evaluate audio-visual understanding, we further propose a benchmark, called AVU-Bench, for AV-LLMs. We highlight the importance of interaction from both modalities, as in Fig. 1 (b).

Our contributions are summarized below:

- We propose Dolphin, a fine-grained AV-LLM for audio-video multimodal and unimodal understanding. Dolphin could remarkably exploit audio information for understanding.
- The core innovation lies in the architecture that enables audio-visual multi-scale spatial alignment as well as context-aided temporal alignment, which ensures fine-grained extraction of two complementary modalities and interaction between them.
- We curate the first large-scale audio-visual caption and instruction-following dataset. It contains 5.2M samples with several splits and does not require rigid AV correspondence.
- Extensive experiments show that Dolphin could not only achieve outstanding audio-visual understanding performance, but also be competent in unimodal tasks, which validates that Dolphin effectively exploits the information of two complementary modalities.

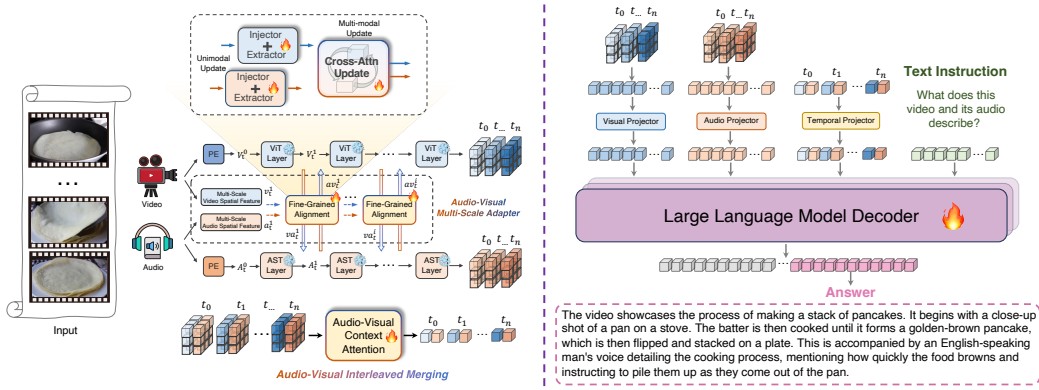

Figure 2: Overview of our Dolphin, which aligns on both spatial and temporal dimensions to fully exploit the natural consistency of videos and enhance the complementary roles of vision and hearing. Specifically, for spatial alignment, we introduced an audio-visual multi-scale adapter using a dual-feature pathway design, which extracts multi-scale features from both visual and auditory inputs and achieves fine-grained alignment with the respective modality.

## 2 RELATED WORKS

**Multi-Modal Large Language Models for Video Understanding.** A series of studies first decompose videos into different representation dimensions and then integrate the inputs to enrich the prompts for MLLMs. For example, Video-ChatGPT (Maaz et al., 2023) splits videos into spatial and temporal branches for pooling. VideoChat (Li et al., 2023b) decomposes videos into descriptions and video embeddings. LLaMA-VID (Li et al., 2023c) represents each frame with context tokens and content tokens. Video-LaVIT (Jin et al., 2024) performs video tokenization using keyframes and motion vectors. Other works incorporate images into unified video training to enrich the training data. Chat-UniVi (Jin et al., 2023) and Video-LLaVA (Lin et al., 2023) employ strategies such as object-based adaptive cluster-based tokens and aligning before projection to unify image and video inputs, thereby achieving more powerful visual understanding.

**Audio-Visual Large Language Models.** Early works, like VideoChat (Li et al., 2023b), simply encode audio inputs using Whisper (Radford et al., 2023b) and directly overlay them to the textual input. Later works seek to align the output of audio and visual encoders before feeding them into LLMs. MACAW-LLM (Lyu et al., 2023) aligns the encoder outputs to the textual space through a learnable alignment module. Audio-Visual LLM (Shu et al., 2023a) activates the embeddings of different modalities with different tags. Moreover, some works (Sun et al., 2023a; Zhang et al., 2023a) explore temporal alignment between video and audio using a Q-Former (Li et al., 2023a) structure, but most of them (Tang et al., 2024) neglect fine-grained spatial modeling. Meerkat (Chowdhury et al., 2024) explores fine-grained understanding but only focuses on images. To sum up, most existing AV-LLMs neither struggle to capture fine-grained local information nor handle temporal alignment, which motivates us to delve into the design and training of AV-LLMs.

## 3 MODEL ARCHITECTURE

**Overview.** Dolphin primarily focuses on effectively strengthening the fine-grained alignment and interaction between visual and auditory modalities. It effectively exploits the complementary information of two modalities and prevents overlooking any of them. Specifically, Dolphin primarily comprises three components: (1) Audio-visual (AV) multi-scale adapter (Section 3.1), which aims to align audio and visual features across various spatial scales in a fine-grained manner. (2) Audio-visual (AV) interleaved merging (Section 3.2) for temporal alignment and extracting complementary information of two modalities. (3) Large Language Model (LLM) to handle the interacted audio-visual tokens and output responses according to the instructions.

**Notations.** We split each video into $T = 8$ visual frames and audio clips. Let $H_v, W_v$ denote the height and width of each frame, while each audio clip is transformed into a spectrogram of $H_a \times W_a$. $N$ denotes the number of ViT blocks. In the AV multi-scale adapter, the visual modality

contains dual-pathway input features, *i.e.*, global feature $\mathcal{V}_t^i$ and multi-scale local feature $v_t^i$, where $i$ denotes the $i$-th block and $t$ denotes the $t$-th frame. Besides, $\widehat{\mathcal{V}}_t^i$ and $\hat{v}_t^i$ are features after modality interaction. Similar expressions of $\mathcal{A}_t^i, a_t^i, \widehat{\mathcal{A}}_t^i, \hat{a}_t^i$ are applicable to audio. The joint feature after modality interaction could be represented as $av_t^i$. In AV interleaved merging, visual and audio tokens after contextual interaction could be written as $\mathcal{V}_t^{temp}$ and $\mathcal{A}_t^{temp}$, respectively.

## 3.1 SPATIAL: AUDIO-VISUAL MULTI-SCALE ADAPTER

The AV multi-scale adapter is designed to enhance the fine-grained alignment of spatial features. Taking the spirit of ViT-Adapter (Chen et al., 2023b), we first independently extract each modality's global and pyramid multi-scale features Lin et al. (2017); Liu et al. (2025). Then, fine-grained alignment is performed across different scales. In this section, we introduce the data flow with the visual modality as an example, and the same principle applies to audio as well.

**Visual global and initial multi-scale features.** For ViT-L (Dosovitskiy et al., 2021), we divide it into $N = 4$ blocks, each with 6 layers. To acquire multi-scale spatial features, we feed the images into the spatial module, *i.e.*, pyramid convolutional network (Lin et al., 2017), to extract various multi-resolution features, *i.e.*, 1/8, 1/16 and 1/32 of $H_v \times W_v$, forming the initial $D$-dimensional multi-scale features $v_t^1$ as an input to the adapter, as shown below:

$$v_t^1 \in \mathbb{R}^{\left(\frac{HW}{8^2} + \frac{HW}{16^2} + \frac{HW}{32^2}\right) \times D}, \quad a_t^1 \in \mathbb{R}^{\left(\frac{HW}{8^2} + \frac{HW}{16^2} + \frac{HW}{32^2}\right) \times D}. \tag{1}$$

**Inter-modality feature interaction.** To achieve fine-grained alignment across scales, audio features are injected into the multi-scale visual features $v_t^i$, where $i$ denotes the $i$-th block, we incorporate audio global feature $\mathcal{A}_t^i$ into $v_t^i$ through cross-attention, and obtain audio-guided multi-scale visual feature $av_t^i$ as follows:

$$av_t^i = \mathrm{CrossAttn}(v_t^i, \mathcal{A}_t^i, \mathcal{A}_t^i). \tag{2}$$

**Intra-modality feature fusion.** As shown in Figure 2, in the fusion process, the audio-guided multi-scale visual features $av_t^i$, which contains visual spatial priors and audio information, are injected into original global features $\mathcal{V}_t^i$ through cross-attention and added to $\mathcal{V}_t^i$, as follows:

$$\widehat{\mathcal{V}}_t^i = \mathcal{V}_t^i + \beta^i \, \mathrm{CrossAttn}(\mathcal{V}_t^i, av_t^i, av_t^i). \tag{3}$$

Here, $\beta$ is a learnable vector initialized as 0, ensuring the initial outputs are the same as ViT's global features $\mathcal{V}_t^i$. Subsequently, the global features $\widehat{\mathcal{V}}_t^i$ are transmitted to the next block through the $i$-th standard ViT block and the fine-grained features $\hat{v}_t^i$ are passed to the next block as $v_t^{i+1}$ through cross-attention and FFN layers:

$$\mathcal{V}_t^{i+1} = \mathrm{ViT\text{-}Block}^i(\widehat{\mathcal{V}}_t^i) \tag{4}$$

$$v_t^{i+1} = \hat{v}_t^i + \mathrm{FFN}\left(\hat{v}_t^i\right), \quad \hat{v}_t^i = v_t^i + \mathrm{CrossAttn}\left(v_t^i, \mathcal{V}_t^{i+1}, \mathcal{V}_t^{i+1}\right). \tag{5}$$

**Final outputs.** The AV multi-scale adapter obtains $\mathcal{V}_t^{N+1} \in \mathbb{R}^{B \times T \times L_v \times D}$ and $\mathcal{A}_t^{N+1} \in \mathbb{R}^{B \times T \times L_a \times D}$, where $B, T, D$ denote batch size, number of frames and feature dimension, and $L_v, L_a$ are the number of tokens of two modalities. We perform average pooling across the temporal dimension $T$ and obtain the final spatial tokens. In this way, we gradually incorporate audio information into the visual features and enhance the interaction between audio and visual modalities. The guidance provided by audio to the multi-scale visual features facilitates fine-grained alignment.

## 3.2 TEMPORAL: AUDIO-VISUAL INTERLEAVED MERGING

In this stage, we merge the final audio and visual features to implement alignment and interaction at the temporal axis. Specifically, by concatenating visual and audio tokens in the same frame, we could obtain $T$ pairs of audio-visual interleaved tokens, each referred to as an AV group, as shown in Figure 2. Within each group, we perform bi-directional contextual attention on visual and audio tokens, which produce visual-contextualized audio tokens and audio-contextualized visual tokens.

$$\mathcal{V}_t^{temp} = \mathrm{CrossAttn}(\mathcal{A}_t, \mathcal{V}_t, \mathcal{V}_t), \quad \mathcal{A}_t^{temp} = \mathrm{CrossAttn}(\mathcal{V}_t, \mathcal{A}_t, \mathcal{A}_t). \tag{6}$$

Then each AV token group is mapped into the input space of LLM through a joint audio-visual projector. Here, we condense each frame of the video into two tokens. In this way, we achieve integration and merging of audio and visual information, which enhances the audio-visual information exploitation of AV-LLMs.

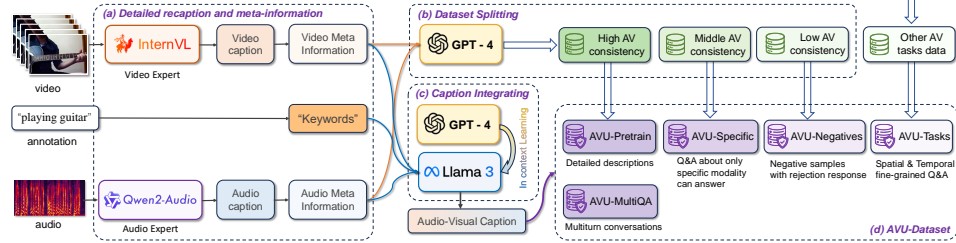

Figure 3: The integration pipeline of the audio-visual understanding dataset (AVU-dataset).

Table 1: The detailed statistics for our AVU (Audio-Visual Understanding Dataset).

| Subsets | AVU-Pretrain | AVU-MultiQA | AVU-Specific | | AVU-Negetive | AVU-Tasks | |
|---|---|---|---|---|---|---|---|
| Instruction Type | Detailed Description | Multiturn Conversation and Reasoning | Visual-Specific Info | Audio-Specific Info | Negative Samples | Temporal and Spatial | Total |
| Audio-Video Statistics | 1.11M | 500k (subset as AVU-Pretrain) | 554k | | 186k | 283k | 2.13M |
| Instruction Statistics | 1.11M | 1M | 1.09M | 1.11M | 370k | 559k | 5.24M |

## 3.3 TRAINING STRATEGY

We employ Vicuna-v1.5 (Chiang et al., 2023) as the LLM. The video encoder is ViT-L/14 from CLIP (Radford et al., 2021), and the audio encoder adopts ImageBind (Girdhar et al., 2023b). (1) In the initial pre-training phase, we freeze the visual encoder, audio encoder, and LLM and only update all the projectors as well as the AV multi-scale adapter to achieve alignment across visual, auditory, and LLM modal spaces. We use the AVU-Pretrain dataset. (2) For the instruction tuning phase, only the visual and audio encoders are frozen, while other modules are updated. We employ AVU-Multi Q&A, AVU-Specific, AVU-Negatives and AVU-Tasks subsets. See Section 4.3 for more details.

## 4 AVU: AUDIO-VISUAL UNDERSTANDING DATASET

**Motivation.** The quality of data (Ma et al., 2024b; 2025b) plays a vital role in performance. Currently, there is a shortage of large-scale audio-visual instruction-following and fine-grained captions, which hinders the model from focusing on modality-specific information and potentially leads to audio hallucinations. To solve the issues, we propose an audio-visual understanding (AVU)-dataset, a large-scale AV understanding and instruction-following dataset.

**Overview.** In this section, we introduce the construction of the AVU-dataset (Figure 3). Specifically, we first re-caption (Section 4.1) audio and video data and generate meta-information (Section 4.2). Then the dataset is split (Section 4.3) into several subsets according to the similarity between audio and visual meta-information, and we feed different prompt templates to generate the corresponding instruction-following dataset and training stages.

**Dataset statistics.** As shown in Table 1, AVU-dataset contains 2.13M audio-video pairs, each has several Q&A pairs, resulting in 5.24M Q&A pairs in total. AVU-dataset has four subsets, *i.e.*, AVU-Pretrain (1.11M samples and Q&A), AVU-Multi Q&A (500k samples and 1M Q&A pairs), AVU-Specific (554K samples and 1.09M Q&A pairs for video and 1.11M for audio), AVU-Negative (186Ksamples and 370K Q&A pairs) and AVU-Tasks (283K samples and 559K Q&A pairs).

**Datasets quality and verification.** We design three types of filtering mechanisms, including CLIP-Score filtering, Self-consistency filtering, and Annotation filtering, to filter noisy samples and guarantee the dataset quality, and then human verification is implemented. Details are shown in the Appendix B.

### 4.1 DETAILED RE-CAPTION GENERATION

**Source datasets.** We collect widely used audio-visual datasets, including AudioSet-2M (Gemmeke et al., 2017), VGG-Sound (Chen et al., 2020a) and task-specific datasets, *i.e.*, MUSIC (Li et al., 2022) for AVQA (Yang et al., 2022), Flickr-SoundNet (Arandjelovic & Zisserman, 2017), VGG-SoundSource for AV source localization (Mo & Morgado, 2022), AVE dataset for AV event

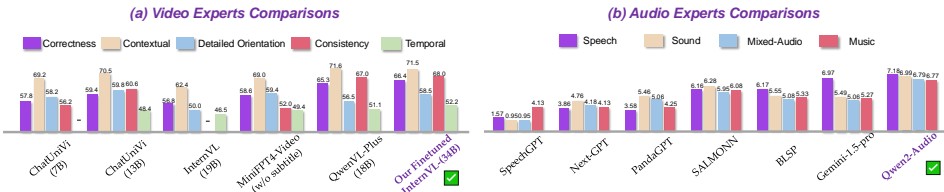

Figure 4: Performance comparison of different task-specific experts.

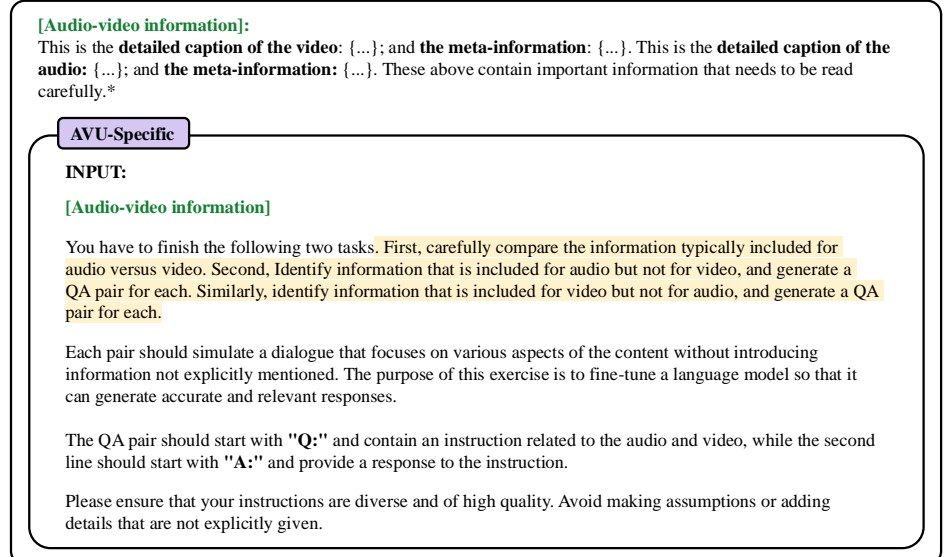

Figure 5: Examples of prompt templates for generating AVU-dataset, others are in the appendix.

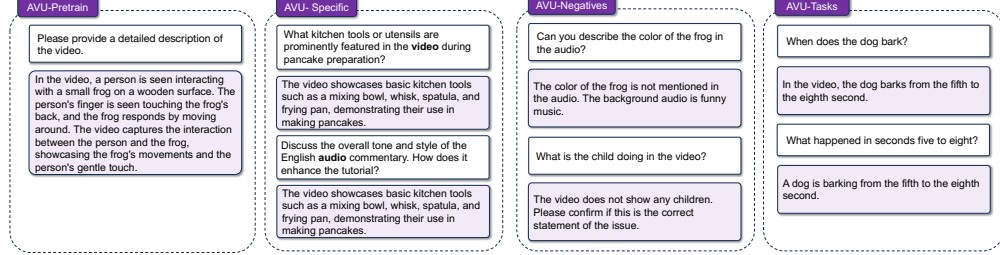

Figure 6: Some examples of AVU-dataset.

localization (Tian et al., 2018), AVS-dataset for AV segmentation (Zhou et al., 2022), and LLP for AV video parsing (Tian et al., 2020). Then we utilize expert models to re-caption audio and video to obtain audio, video and audio-video captions.

**Expert models.** For MLLM, we employ a fine-tuned version of InternVL-34B (Chen et al., 2024). For the audio expert captioners, we choose Qwen2-Audio-7B-Instruct (Chu et al., 2024). The performance of these models is illustrated in Figure 4.

**Prompt templates.** We prompt expert models with hand-crafted templates to generate detailed captions, and set some regularizations to mitigate hallucinations. Figure 5 displays a subset: AVU-Specific prompt templates. See Figure 9 and Figure 10 in the appendix for more details.

## 4.2 META-INFORMATION GENERATION AND INTEGRATION

**Meta-information generation.** The function of meta-information is to maintain audio and video details when integrating unimodal audio and video captions into multimodal audio-video captions. Meta-information could be utilized to guide subsequent AV caption integration and dataset splitting. Specifically, we design options of 'event', 'object', 'scene', 'place', 'action' and 'emotion', *etc*,

and transform original annotations to 'keywords'. Then, we employ GPT-4 (Achiam et al., 2023) to judge consistency based on keywords and meta-information and filter noisy samples, which might be caused by experts' bias and hallucinations.

**Meta-information integration.** We choose LLaMA3-70B-Instruct (Dubey et al., 2024) to integrate meta-information. First, we employ GPT-4 to generate multiple examples of integrating keywords and meta-information. After human adjustments, we feed them to LLaMA3. The in-context learning abilities enable the generation of audio-visual captions.

### 4.3 DATASET SPLITS

**Splitting process.** The dataset is split according to the consistency between audio and video meta-information. **Notably, we do not require strict audio-visual consistency, which is the main novelty of this work**. For different types of data, we create various types of instruction-tuning datasets for corresponding training stages. The whole dataset is divided into three subsets based on audio-visual consistency. Specifically, we feed audio and video meta-information to GPT-4 (Achiam et al., 2023) to obtain the consistency score. Figure 6 shows examples of subsets of the AVU-dataset.

**AVU-Pretrain** comprises samples with high AV consistency. The audio and video information are nearly the same. These samples are suitable for the pre-training stage to align AV modalities. We design fixed question templates (Figure 9 (a)) and randomly select one each time, and use the previously integrated AV captions as the answers. In this way, we obtain AVU-Pretrain subsets.

**AVU-Multi Q&A** also consists of high consistency samples. Different from AVU-Pretrain, we design templates (Figure 9 (b)) to transfer AV caption to multi-turn Q&A and reasoning.

**AVU-Specific** comprises samples with medium AV consistency. Both audio and video carry relatively additional information compared to each other. Questions are posed regarding this additional information to generate Q&A pairs. These Q&A pairs construct AVU-Specific subsets and could only be answered by focusing on a specific modality (Figure 10 (c)).

**AVU-Negatives** consist of low-consistency samples, *e.g.*, the sounding object is not present in the frame. Taking the spirit of contrastive learning (Chen et al., 2020b; He et al., 2020), we create the negative sample dataset, *i.e.*, AVU-Negatives (Figure 10 (d)), whose answers are primarily used as rejection. This subset could teach LLMs the rejection option, mitigating potential hallucinations.

**AVU-Tasks** are curated directly from downstream AV tasks. Specifically, we transform the original annotations to the format of Q&A, including accurate details like time, spatial, and event. Notably, the AVU-Tasks are derived from the fine-grained annotations, and significantly contribute to the model's fine-grained alignment capabilities.

For pre-training, we mainly employ high-consistency samples to enhance modality alignment, while for instruction-tuning, we make use of samples in which audio and visual do not exactly overlap and mix AVU-Multi Q&A, AVU-Specific, AVU-Negatives and AVU-Tasks for training. In this way, the issue of neglect of audio and hallucination is mitigated.

## 5 EXPERIMENTS

We introduce the experimental setup and comparisons among models. In the ablation studies, we explored and validated that fine-grained alignment significantly aids LLMs in multimodal understanding. Temporal contextual alignment is an effective way to leverage the inherent consistency of videos and to exploit complementary audio-visual information. Additionally, we also conducted ablations on the proposed dataset, showcasing its assistance in audio-visual joint perception and its high quality.

### 5.1 EXPERIMENTAL SETUP

**Implementation details.** For each video, we extract 8 frames of $224 \times 224$ resolution, and audio is sampled into 8 frames, each turned into a $128 \times 204$ spectrogram. Both pre-training and fine-tuning are conducted for one epoch, with batch sizes of 256 for pretraining and 128 for finetuning. Projectors for audio, video, and audio-video use two-layer MLPs with a GELU (Hendrycks & Gimpel, 2016) activation. Training is performed on NVIDIA A100 GPUs. More details are in the appendix.

Table 2: Comparison with existing Video-LLMs. We conducted a performance comparison with the existing Video-LLM and reported the scoring results of GPT on four zero-shot video-QA datasets.

| Method | LLM | MSVD-QA Acc | MSVD-QA Score | MSR-VTT-QA Acc | MSR-VTT-QA Score | ActivityNet-QA Acc | ActivityNet-QA Score | TGIF Acc | TGIF Score | POPE |
|---|---|---|---|---|---|---|---|---|---|---|
| LLaMA-Adapter (Zhang et al., 2023b) | LLaMA-7B | 54.9 | 3.1 | 43.8 | 2.7 | 34.2 | 2.7 | - | - | - |
| Video-Chat (Li et al., 2023b) | Vicuna-7B | 56.3 | 2.8 | 45.0 | 2.5 | 26.5 | 2.2 | - | - | - |
| Video-ChatGPT (Maaz et al., 2023) | Vicuna-7B | 64.9 | 3.3 | 49.3 | 2.8 | 35.2 | 2.7 | - | - | - |
| BT-Adapter (Liu et al., 2024) | Vicuna-7B | 67.5 | 3.7 | 47.0 | 3.2 | 45.7 | 3.2 | - | - | - |
| LLaMA-VID (Li et al., 2023c) | Vicuna-7B | 69.7 | 3.7 | 57.7 | 3.2 | 47.4 | 3.3 | - | - | - |
| LLaMA-VID (Li et al., 2023c) | Vicuna-13B | 70.0 | 3.7 | 58.9 | 3.3 | 47.5 | 3.3 | - | - | - |
| Video-LLaVA (Lin et al., 2023) | Vicuna-7B | 70.7 | **3.9** | 59.2 | **3.5** | 45.3 | 3.3 | 70.0 | **4.0** | 84.4 |
| PandaGPT (Sun et al., 2024) | Vicuna-7B | 46.7 | - | 23.7 | - | - | - | - | - | 78.5 |
| VideoLLaMA (Zhang et al., 2023a) | Vicuna-7B | 51.6 | 2.5 | 29.6 | 1.8 | 12.4 | 1.1 | - | - | 82.9 |
| AV-LLM (Shu et al., 2023a) | Vicuna-7B | 67.3 | - | 53.7 | - | 47.2 | - | - | - | - |
| AVicuna (Tang et al., 2024) | Vicuna-7B | 70.2 | - | 59.7 | - | **53.0** | - | - | - | 84.1 |
| OneLLM (Han et al., 2024) | LLaMA2 | 56.5 | - | 53.8 | - | - | - | - | - | - |
| FAVOR (Sun et al., 2023b) | Vicuna-7B | 67.8 | - | 59.3 | - | - | - | - | - | - |
| VideoLLaMA 2 (Cheng et al., 2024) | Mistral-Instruct | 71.7 | **3.9** | 57.4 | - | 49.9 | 3.3 | - | - | 85.4 |
| video-SALMONN (Sun et al., 2024) | Vicuna-7B | 67.9 | 3.7 | 59.5 | 3.4 | - | - | - | - | - |
| Dolphin-7B-LoRA (Ours) | Vicuna-7B | 71.6 | **3.9** | 61.3 | 3.4 | 47.9 | **3.4** | 70.9 | 3.9 | 85.1 |
| Dolphin-7B (Ours) | Vicuna-7B | 72.7 | **3.9** | 62.6 | 3.5 | 49.1 | **3.4** | 71.2 | 3.9 | 85.9 |
| Dolphin-13B (Ours) | Vicuna-13B | **74.8** | **3.9** | **63.5** | **3.5** | 49.6 | **3.4** | **71.3** | **4.0** | **86.2** |

Table 3: Comparison with Audio-LLMs. We conducted closed-ended and open-ended auditory tasks with LTU and LTU-AS, where ZS denotes zero-shot evaluation.

| Method | Audio Classif. ESC-50 (ACC↑) | Audio Caption AudioCaps (SPICE↑) | Speech Recognition Librispeech (WER↓) | Emotion Recognition IEMOCAP (ACC↑) | Gender Classif. Voxceleb2 (maro-F1↑) | Age Pred. Voxceleb2 (MAE↓) | Music Genre Classif. GTZAN (ACC↑) | Audio Question (ACC↑) | Speech Question (ACC↑) | Audio Hallucination Random (ACC↑) |
|---|---|---|---|---|---|---|---|---|---|---|
| *Best specialized models trained supervisedly on each dataset. Not generalizable to unseen label sets and tasks.* | | | | | | | | | | |
| TASK-SOTA | 97.0 | 17.7 | 1.4 | 70.6 | 98.3 | 9.4 | - | - | - | |
| *CLIP-like audio-text model.* | | | | | | | | | | |
| AudioClip (Guzhov et al., 2022) | 69.4 | - | - | - | - | - | - | - | - | - |
| CLAP (Huang et al., 2013) | 82.6 | - | - | - | - | - | 25.2 | - | - | - |
| LTU-Audio (Gong et al., 2023b) | 82.8 | 17.0 | 104.2 | 38.2 | 77.0 | - | 29.8 | 96% | 69% | - |
| LTU-Speech (Gong et al., 2023b) | 10.9 | 0.5 | 12.9 | 69.8 | 90.1 | 7.9 | 23.5 | 65% | 93% | - |
| LTU-AS (Gong et al., 2023b) | 80.8[zs] | 15.0 | 4.9 | 65.2 | 90.8 | 7.3 | 50.3[zs] | 96% | 94% | 50.1 |
| VideoLLaMA (Zhang et al., 2023a) | 62.6[zs] | 6.2 | 128.4[zs] | 23.4[zs] | 43.5[zs] | 8.8 | 22.2[zs] | 56% | 27% | 43.2 |
| VideoLLaMA 2 (Cheng et al., 2024) | 74.8[zs] | 15.8 | 9.8[zs] | 63.9[zs] | 89.7[zs] | 7.3 | 34.9[zs] | 92% | 91% | 56.8 |
| video-SALMONN (Sun et al., 2024) | 77.6[zs] | 16.6 | 3.9[zs] | 65.5[zs] | 90.6[zs] | 7.4 | 36.7[zs] | 95% | 94% | 58.2 |
| Dolphin-LoRA (Ours) | 81.6[zs] | 17.2 | 12.8[zs] | 67.4[zs] | 91.2[zs] | 7.2[zs] | 33.6 | 96% | 93% | 58.6 |
| Dolphin (Ours) | 83.1[zs] | 17.8 | 8.3[zs] | 69.2[zs] | 92.5[zs] | 7.0[zs] | 37.8 | 96% | 94% | 63.2 |

**Dataset.** During training, we enhanced our model by mixing inputs from multiple modalities. Apart from using AVU, we employ LLaVA (Liu et al., 2023), 10% of Valley, and audio clips for pre-training, LLaVA_instruct, Video-ChatGPT (Maaz et al., 2023), and ClothoV2 (Drossos et al., 2020) for instruction-based finetuning. We assessed our model's zero-shot capabilities on single-modality video and audio Q&A benchmarks and compared it against existing audio-visual LLMs using a tailored downstream task benchmark.

## 5.2 COMPARISON WITH STATE-OF-THE-ARTS

**Zero-Shot Video Understanding.** To validate our model's efficacy, we compared its performance in video comprehension with existing video-LLMs. Specifically, we showcased Dolphin's comparison with various methods on MSR-VTT-QA, MSVD-QA, and ActivityNet-QA benchmarks. As indicated in Table 2, our model demonstrated superior video understanding, proving that auditory information complemented visual comprehension in the training stage. Moreover, POPE (Li et al., 2023d) results show our method could mitigate object hallucinations.

**Closed and Open-Ended Audio Tasks.** To test our model's audio understanding, we followed LTU and LTU-AS, evaluating closed-ended and open-ended (Ma et al., 2024a; Zhu et al., 2024) audio tasks like classification and captioning in Table 3. Despite zero-shot settings on many datasets, our model showed effective understanding, achieving or exceeding SOTA performance. We use the pre-trained ImageBind (Girdhar et al., 2023a) audio encoder as our audio encoder and freeze it during training, whose structure is AST (also used in LTU). It is less effective in speech recognition compared with

Table 4: Results on the proposed audio-visual understanding bench and AVSD.

| Task | AVU | | AVSL | | AVE | | AVVP | | AVSD |
|---|---|---|---|---|---|---|---|---|---|
| Dataset | MUSIC | | AVL, AVS | | AVE | | LLP | | AVSD |
| Metric | acc | score | acc | score | acc | score | acc | score | acc |
| Video LLaMA (Zhang et al., 2023a) | 65.4 | 3.3 | 35.6 | 2.3 | 40.8 | 2.8 | 21.5 | 2.1 | 36.7 |
| Video LLaMA 2 (Cheng et al., 2024) | 73.2 | 3.5 | 47.1 | 2.3 | 48.2 | 2.9 | 28.4 | 2.3 | 57.2 |
| video-SALMONN (Sun et al., 2024) | 74.7 | 3.5 | 48.3 | 2.4 | 51.5 | 3.0 | 30.9 | 2.4 | 51.6 |
| Dolphin (Ours) | **78.2** | **3.9** | **51.8** | **3.0** | **52.1** | **3.2** | **31.4** | **2.8** | **59.1** |

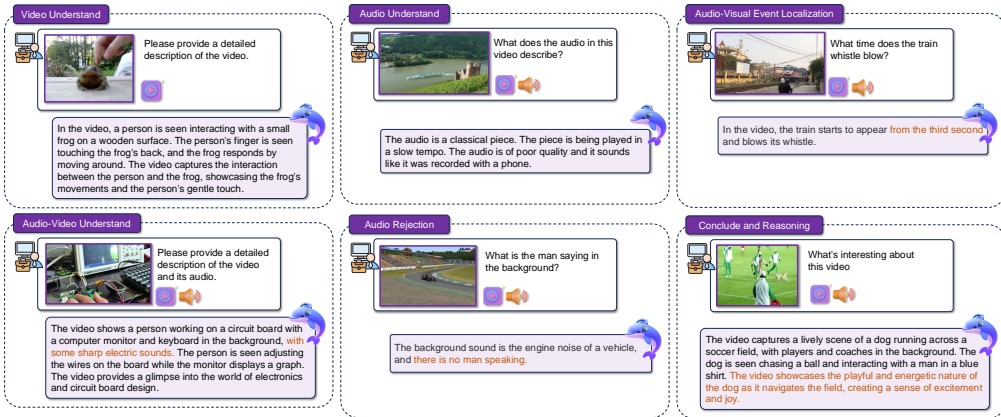

Figure 7: Qualitative cases of Dolphin.

Whisper (used by LTU-AS). However, our model showed improved speech recognition, benefiting from our dataset mixing audio and speech. Additionally, the audio-hallucination (Kuan et al., 2024) experiment shows that our model can effectively reduce audio hallucination.

**Audio-Visual Understanding Bench.** To assess our model's competency in audio-visual understanding, we developed a minibench tailored for Large Language Models (LLMs) that aligns with existing audio-visual tasks. We meticulously gathered test datasets from labeled audio-visual tasks, comprising MUSIC (AVQA), LLP (AVVP), AVE (AVE), AVS-Bench (AVS), Flickr-SoundNet, and VGG-SoundSource (AVL), to ensure a fair and precise evaluation. Inspired by zero-shot question-answer evaluations and Video-ChatGPT, we transformed these tasks' ground truths into open-ended question-answer formats. We also evaluate our model on AVSD (Alamri et al., 2019) benchmark.

This method evaluates the precision of our model's predictive output, awarding scores on a 1-5 scale. Comparing the performance of existing methods, like Video-LLaMA (Zhang et al., 2023a), Video-LLaMA 2 (Cheng et al., 2024), and video-SALMONN (Sun et al., 2024), our results, presented in Table 4, demonstrate a notable performance gap favoring our model on audio-centric vision tasks, which highlights our model's enhanced comprehension of audio and visual modalities.

**Qualitative evaluation.** The qualitative cases of the proposed Dolphin are illustrated in Figure 7. Results show that Dolphin could remarkably comprehend both audio and visual modalities, together with enhanced video understanding.

## 5.3 ABLATION STUDIES AND ANALYSIS

In this section, through detailed ablation studies, we explored how to enhance an audio-visual LLM's integration of video and audio for better video understanding. We also validated the effectiveness of our proposed methods and datasets through model and dataset ablations.

**Fine-grained spatial alignment effectively aids AV-LLMs in understanding multimodal semantic information.** In our prior analysis, we found that without fine-grained interaction between visual and auditory information, LLMs struggle to learn relevant information between videos and audios, as the lack of prior knowledge regarding the two modalities leads the model to focus more on the information-rich video content while neglecting audio. To investigate this issue, we conducted ablation studies on our fine-grained alignment module in Table 5a. The results reveal that inter-

modality interaction indeed enhances the LLM's attention to both modalities. However, in comparison, when the two modalities undergo fine-grained alignment before projection, the model is not only able to simultaneously focus on both modalities but also better extract the complementary information from visual and audio. Table 5a illustrates that fine-grained audio-visual alignment aids LLMs in cross-modal understanding. Inter-modal interaction boosts attention to both modalities, but fine-grained alignment before projection further allows simultaneous focus and better extraction of visual and auditory complementary information. Compared to removing temporal alignment, understanding of videos significantly improves with the help of temporal alignment and interaction.

**Contextual alignment effectively leverages the inherent consistency of videos.** We analyzed our temporal context attention module in Table 5a and found that models without it or with only the chronological arrangement of tokens underperform. In contrast, employing temporal context attention, which aligns video and audio features over time, significantly boosts performance by tapping into their inherent temporal consistency, thus enhancing LLM's understanding of both modalities together.

Table 5: Ablations on model architecture designs (a) and various dataset subsets (b).

| Module | Variants | AVU | | AVE | |
|---|---|---|---|---|---|
| | | acc | score | acc | score |
| Original | Dolphin (All) | **78.2** | **3.9** | **52.1** | **3.2** |
| Spatial | w/o inter-modal | 77.6 | 3.8 | 51.8 | 3.2 |
| | w/o AV adapter | 75.4 | 3.6 | 51.6 | 3.2 |
| Temporal | w/o bi-dir context attn | 76.1 | 3.6 | 50.3 | 3.1 |
| | w/o AV inter-merging | 32.3 | 2.5 | 22.6 | 2.2 |

(a) Model architecture designs.

| Variants | MSR-VTT-QA | | AV Understand | | POPE |
|---|---|---|---|---|---|
| | acc | score | acc | score | |
| w/o AVU-Pretrain | 58.8 | 3.3 | 69.8 | 3.6 | 81.7 |
| w/o AVU-Specific | 59.3 | 3.4 | 72.6 | 3.6 | 82.1 |
| w/o AVU-Negatives | 61.0 | 3.5 | 77.8 | 3.8 | 84.8 |
| Full AVU | **62.6** | **3.5** | **78.2** | **3.9** | **85.9** |

(b) AVU-dataset subsets.

Table 6: Detailed ablation on datasets and models.

| Methods | MSR-VTT-QA | | Audiocaps | AV Understand | |
|---|---|---|---|---|---|
| | acc | score | SPICE | acc | score |
| Video-LLaMA + Video-LLaMA dataset | 29.6 | 1.8 | 11.8 | 65.4 | 3.3 |
| Video-LLaMA + AVU (Ours) | 55.3 (+25.7) | 3.1 (+1.3) | 15.9 (+4.1) | 73.2 (+7.8) | 3.6 (+0.3) |
| Dolphin + Video-LLaMA dataset | 42.8 (+13.2) | 2.7 (+0.9) | 14.4 (+2.6) | 69.9 (+4.5) | 3.4 (+0.1) |
| Dolphin + AVU (Ours) | 62.6 (+19.8) | 3.4 (+1.6) | 17.8 (+3.4) | 78.2 (+6.3) | 3.9 (+0.5) |

**Effectiveness of model structure and dataset quality.** We conducted an ablation study on our model and dataset in Table 5b. First, training without the audio dataset diminished the model's understanding of pure video content, indicating that our model effectively utilizes complementary information from audio. Moreover, comparing the performance without our dataset and training Video-LLaMA with our dataset (Table 6) showed a significant performance decline without the AVU dataset, whereas Video-LLaMA achieved better performance on our dataset. These result, along with the human validation in Appendix.B, jointly demonstrates the quality and effectiveness of our dataset.

## 6 CONCLUSION

In this paper, we primarily explored how existing AV-LLMs can effectively overcome insufficient focus on audio information. We innovatively propose Dolphin, an Audio-Visual Large Language Model for audio-video understanding, which features fine-grained interaction and alignment on both spatial and temporal levels. We design a multi-scale audio-visual adapter and a temporal context module to fully leverage the inherent consistency of videos and realize the complementary function of visual and auditory information. Extensive experiments indicate the effectiveness of the model. Additionally, we collected and labeled an audio-visual caption and instruction fine-tuning dataset for video understanding, containing 2.13M pairs of AV samples and 5.24M Q&A pairs and providing diverse training objectives for audio-visual LLMs. This instruction-following dataset is suitable for large model learning and has been proven to effectively enhance the audio-visual understanding capabilities of existing models, and demonstrate the quality of our dataset. Finally, through experiments, we explore and conclude that fine-grained temporal and spatial alignment can effectively enhance visual and auditory perception abilities of audio-visual LLMs.

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

# Supplementary Material

## A  DIFFERENCES FROM EXISTING METHODS

### A.1  COMPARISON WITH EXISTING AV-MLLMS.

Here, we also explain the differences from the methodology perspective. MACAW-LLM: The training visual and audio signals come from different videos, which results in a lack of fine-grained representation and alignment of modalities. The proposed dataset includes only images and videos, without any audio.

**ImageBind-LLM** (Han et al., 2023): It includes six modalities but only utilizes image-text alignment for training, without specifically addressing the alignment and representation of audio-video pairs.

**PandaGPT** (Su et al., 2023): It integrates a shared latent space derived from ImageBind, primarily facilitating zero-shot transfer across six modalities: text, image/video, audio, depth, thermal, and IMUs.

**Video-LLaMA** (Zhang et al., 2023a): It employs audio and visual Q-Former for respective modalities, but it only trains the vision-language branch and the A-L branch on video/image instruction data, without incorporating audio training.

**AV-LLM** (Shu et al., 2023a): It focuses solely on spatiotemporal modeling of the video modality, neglecting the fine-grained information from the audio.

**FAVOR** (Sun et al., 2023a): It proposes a causal Q-Former structure with a causal attention module that aligns only temporally, lacking fine-grained audio-visual modeling.

**Video-LLaMA2** (Cheng et al., 2024): It focuses solely on the spatiotemporal representation of video, neglecting fine-grained audio representation and audio-visual interactions.

**Meerkat** (Chowdhury et al., 2024): It is a fine-grained audio-visual understanding model; however, it only models images and lacks video understanding capabilities.

**AVicuna** (Tang et al., 2024): It focuses solely on temporal modeling, neglecting spatial fine-grained information. Additionally, it exhibits a significant amount of hallucination.

**OneLLM** (Han et al., 2024): It achieves effective integration and instruction adherence across different modalities through progressive multimodal alignment and modality routing, primarily focusing on global alignment.

In summary, our work introduces innovations in both the framework and dataset pipeline, effectively addressing the challenges faced by AV-MLLMs. Furthermore, the proposed solutions have the potential to inspire research on existing AV-MLLM models and datasets, while also contributing to the broader MLLM community.

## A.2 COMPARISON BETWEEN DOLPHIN AND LTU AND LTU-AS.

We summarize the following four differences: (1) Different model types. LTU (Gong et al., 2023b) and LTU-AS (Gong et al., 2023a) are audio/speech-specific models, trained specifically with audio/speech-language models for audio or speech tasks. In contrast, our Dolphin model is an AV-LLM that comprehends both audio and video, encompassing a broader range of modalities. (2) Different audio backbones. LTU-AS employs Whisper (Radford et al., 2023b), which is pre-trained on a large-scale speech-language dataset, resulting in stronger speech recognition performance. We use the ImageBind-aud encoder (used by LTU), which has not been pre-trained on a speech dataset. Moreover, the primary tasks in Table 3, such as speech recognition, emotion recognition, and gender classification, are closely related to speech. Therefore, our performance reflects zero-shot results. (3) Compared to LTU-AS, our audio encoder's zero-shot recognition performance still achieves state-of-the-art results in the majority of tasks. This demonstrates our model's ability to pay attention to and comprehend audio effectively. (4) Compared to LTU, our performance surpasses theirs by a considerable margin, which can be attributed to our generated dataset that includes both audio and speech samples. This highlights the effectiveness of our dataset.

## B DETAILS OF DATASET CURATION AND VERIFICATION

For the dataset filtering, we design three types of filtering mechanisms: CLIP-Score filtering, self-consistency filtering and annotation filtering. Subsequently, human verification is implemented to quantitatively verify the quality of the generated caption.

- **CLIP-score filtering**. In this stage, the visual and audio experts first generate captions based on the input video and audio, and then we employ CLIP and CLAP to assess the similarity score for each caption, respectively. Captions with lower scores might have noise and hallucinations.

- **Self-consistency filtering**. We further prompt the visual and audio experts to summarize the meta-information of the generated captions, and utilize GPT4 to assign the matching score given the initial caption and its meta-information. Captions with lower scores might have noise and hallucinations.

- **Annotation filtering**. The original annotations of the datasets are transformed to 'keywords'. Then, we employ GPT-4 to judge consistency based on keywords and meta-information and filter noisy samples, which might be caused by experts' bias and hallucinations.

- We synthesized the aforementioned three factors to calculate a weighted confidence score for each sample and subsequently ranked them. The bottom 25% of samples, based on this ranking, will be eliminated. The weights for the screening criteria are as follows: CLIP-Score filtering: 2, Self-consistency filtering: 1, Annotation Filtering: 5.

**Human verification**. After the former three filtering steps, 100 human annotators are employed to give scores (1 to 5) to each of the video-caption and audio-caption pairs, considering completeness and accuracy (related to hallucinations). We randomly sample 100 video-caption and 100 audio-caption pairs in the generated dataset. Besides, we also verify the caption after the integration from two modalities (Integration Effect). The results are shown in Table 7.

Table 7: The mean scoring of human verification on completeness and accuracy of audio (A) and visual (V) modalities.

| Captions | V: Completeness | V: Accuracy | A: Completeness | A: Accuracy | Integration Effect |
|---|---|---|---|---|---|
| Mean scoring | 4.27 | 4.45 | 4.23 | 4.40 | 4.31 |

The verification results show that the generated dataset is precise and accurate.

## C FURTHER IMPLEMENTATION DETAILS

### C.1 TRAINING DETAILS.

Stage-1: Multi-modal text alignment pre-training. The model needs to read audio, video and corresponding textual instructions, which is related to the understanding of audio, video and audio-video contents. The ground truth annotations are the captions of AVU-dataset. Stage-2: Audio-visual dynamic instruction-tuning. The model is required to respond accordingly to various types of instructions. The instructions comprise complex visual, audio and audio-visual understanding tasks. Both the projector and LLM are updated. Dataset prompts, audio or/and video and questions are fed to the model to generate answers. The generated answers are then supervised by the ground truth captions of the datasets to update both the projectors and LLMs. For both stages, the learning objective is autoregressive next-word-prediction loss.

### C.2 THE PROPOSED SPATIAL AND TEMPORAL MODULES ARE MUTUALLY-PROMOTED.

The two modules are utilized to align fine-grained spatial and temporal information for audio-visual data. The AV multi-scale adapter separately extracts multi-scale features from visual and auditory modalities and interacts with the other modality for fine-grained alignment. This promotes fine-grained alignment between audio and video, avoiding the neglect of auditory information and effectively enhancing the model's performance in audio-visual dense prediction tasks. The primary innovation of the temporal interleaved merging lies in simultaneously calculating bidirectional attention for both audio and video features, enabling the alignment of video and audio features in the temporal dimension. This effectively leverages the consistency of videos, improving the LLM's joint understanding of video and audio.

## D FURTHER EXPERIMENTAL RESULTS

### D.1 COMPARISON WITH VARIOUS AUDIO AND VISUAL ENCODERS.

We compared various variants of audio and visual encoders and reported the performance for video/audio/audio-video understanding, as shown in Table 8.

Table 8: Comparison with various audio and visual encoder variants.

| id | Encoders | MSR-VTT-QA | Audio Caption | AV Understand |
|---|---|---|---|---|
| (a) | CLIP+CLAP | 62.5 | 17.3 | 74.2 |
| (b) | ImageBind-video+ImageBind-audio | 58.2 | 15.8 | 72.3 |
| (c) | ImageBind-video+CLAP | 59.8 | 17.6 | 76.5 |
| (d) | CLIP+ImageBind-audio (Ours) | 62.6 | 17.8 | 78.2 |

Then the following conclusions are drawn: (1) If both audio and visual aspects are aligned with language (as in CLIP (Radford et al., 2021), CLAP (Elizalde et al., 2023)), the performance on A/V

Table 9: Comparisons with various visual and LLM backbones.

|  | Audio caption | MSR-VTT-QA | AV Understand |
|---|---|---|---|
| w/ SigLIP (Zhai et al., 2023) | 16.5 | 63.5 | 77.9 |
| w/ Whisper (Radford et al., 2023a) | 17.9 | 61.9 | 77.5 |
| w/ Llama3-8B-Instruct (Dubey et al., 2024) | 18.6 | 64.0 | 80.3 |
| Ours | 17.8 | 62.6 | 78.2 |

understanding tasks tends to be good, but the performance on AV tasks is not high. We attribute this to the fact that the AV encoder, due to lack of alignment, is unable to effectively utilize the AV adapter. (2) Building upon this, we added stage 0.5 before pretraining and pre-trained for one epoch on AudioSet-2M (Gemmeke et al., 2017) using the audio-visual contrastive loss. We observed performance improvement (especially in audio), which validates that fine-grained audio-visual alignment effectively helps the model pay attention to the audio modality. (3) If both audio and video use ImageBind (Girdhar et al., 2023b), which has undergone audio-visual alignment, the A/V/AV understanding capabilities decline. We believe that in Machine Learning Language Models, since the output format is language, using a backbone that has not been aligned with language will affect the final understanding results. (4) If the audio modality is aligned with language (CLAP) and the visual modality is aligned audio-visually (ImageBind-video), the results are inferior compared to the other mode (ImageBind-audio+CLIP). We attribute this to the fact that the semantic density of video is much higher than audio, and video-language alignment can better enhance the model's understanding and instruction-following capabilities.

In summary, we believe that selecting a visual encoder aligned with language and an audio encoder aligned with visual can effectively balance A/V/AV understanding capabilities. Therefore, we chose CLIP+ImageBind as our backbone encoder.

## D.2 OTHER ENCODERS AND LLM ABLATIONS.

The primary objective of our work is to explore effective learning schemes, demonstrating that fine-grained audio-visual alignment can enhance the model's understanding of audio, and mitigate hallucinations. To ensure fair comparisons, we have selected the most widely used encoders with better AV alignment and LLMs.

Moreover, we also include the results with other encoders and LLMs, as shown in Table. 9.

When choosing SigLIP, the visual abilities are enhanced while the audio-related abilities are degraded, because SigLIP has weaker alignment compared with CLIP. When choosing Llama3-8B-Instruct, our method obtains better overall results. We attribute it to the superiority of Llama3 and more parameters (LLaMA3: 8B >Vicuna: 7B).

After replacing the aforementioned backbone, there is an improvement in some results, such as SigLIP's understanding of video and the performance of Llama3-8B-Instruct. This further demonstrates the effectiveness of our framework and dataset. It effectively helps the model focus on both audio and video modalities while also mitigating hallucinations. This also demonstrates the strong generalization capability of our model, allowing it to be applicable to a wider range of backbones.

## D.3 IMPACT OF USING IMAGE AND VIDEO ENCODER

We explore the impact of whether our model uses a video encoder.

Firstly, we observed that preceding video caption models predominantly utilize image encoders (CLIP), e.g., LLaMA-VID(Li et al., 2025) and LLaVA-NeXTVideo (Li et al., 2024), ShareGPT4V (Chen et al., 2023a), Valley (Luo et al., 2023), AV-LLM (Shu et al., 2023b), AVicuna (Tang et al., 2024) and video-SALMONN (Sun et al., 2024), the reason is that CLIP could align better with language and are pre-trained on larger numbers of data with stronger visual abilities.

Additionally, using an image encoder to process video allows for the selection of any frame rate, providing flexibility that aids in modeling temporal information. This is also a similar rationale behind the approach used in VideoLLaMA2 (Cheng et al., 2024).

Table 10: Ablation study of using video encoder and image encoder.

|  | Audio Caption | MSR-VTT-QA | AV Understand |
|---|---|---|---|
| w/ LanguageBind video encoder | 16.4 | 60.8 | 76.4 |
| w/ ImageBind video encoder | 15.8 | 58.2 | 72.3 |
| w/ CLIP image encoder (ours) | **17.8** | **62.6** | **78.2** |

Table 11: Ablation study of freezing or unfreezing encoder.

|  | Audio Caption | MSR-VTT-QA | AV Understand |
|---|---|---|---|
| unfreeze av | 15.3 | 58.7 | 75.9 |
| freeze av (Ours) | 17.8 | 62.6 | 78.2 |

Table 12: Ablation study of freezing or unfreezing encoder.

|  | w/ MLP*2+GELU (Ours) | w/ MLP | w/ Q-Former |
|---|---|---|---|
| MSR-VTT-QA | 62.6 | 60.8 | 58.6 |
| AV Understand | 78.2 | 77.1 | 75.3 |

We also added the results using video encoders in LanguageBind (Zhu et al., 2023) and Image-Bind (Girdhar et al., 2023a), as shown in Table. 10.

The results indicate that replacing the CLIP with other video encoders leads to a decline in performance. This is attributed to CLIP's superior semantic representation capabilities and its alignment with text. Besides, CLIP is pre-trained with a large number of images and has stronger vision abilities.

### D.4 IMPACT OF FREEZE OR UNFREEZE

Moreover, we explored whether freezing has an impact on model performance, and conducted experiments with unfrozen AV encoders, as shown in Table. 11.

From the result, we can see that in the case of unfrozen encoders, the overall results degraded. The increased number of training modules made training more challenging and increased training consumption. The training time was approximately 2.8 times longer than before.

### D.5 IMPACT OF DIFFERENT CONNECTOR

For audio, video, and audio-visual connectors, we follow LLaVA and use a 2x MLP with GELU activation. Regarding the selection of connectors, existing Multimodal Language Learning Models (MLLMs) predominantly opt for relatively simple structures. For instance, LLaVA (Liu et al., 2023) and Video-LLaVA (Lin et al., 2023) utilize MLPs, Cambrian (Tong et al., 2024) employs a Spatial Vision Aggregator, and MM1.5 (McKinzie et al., 2025; Zhang et al., 2024) selects an abstractor. An empirical study also indicates that the choice of simple connectors (such as C-Abstractor, average pooling, and attention pooling) has a marginal impact on the results. Consequently, we followed LLaVA and opted for the simplest configuration of MLP $2\times$ with GELU activation.

We found that video-SALMONN (Sun et al., 2024), and VideoLLaMA (Zhang et al., 2023a) use Q-Former as a connector. Therefore, we chose Q-Former and a one-layer MLP to conduct the connector ablation experiments.

From the Table. 12, we can observe that the performance of our model declined slightly after adding the Q-Former, which might be due to the complexity of training the Q-Former, making it difficult to train well. video-SALMONN (Sun et al., 2024), and VideoLLaMA (Zhang et al., 2023a) mainly explore speech and audio with variable resolution and length, while our main concern is the fine-grained audio-visual alignment and mitigation of hallucination. Therefore, we decided not to adopt the Q-Former.

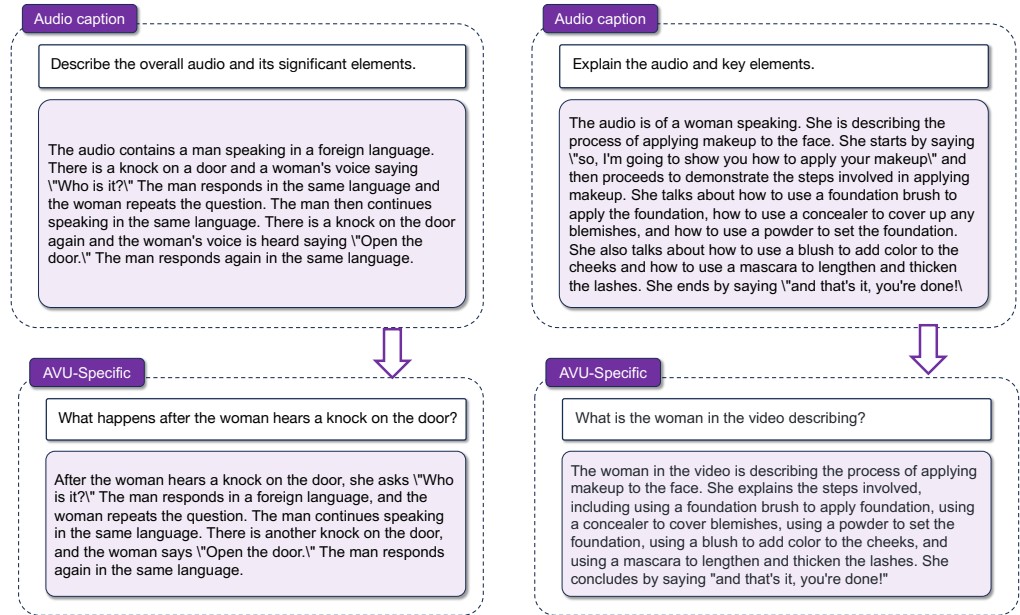

Figure 8: Some cases of dataset samples containing speech information. On the left is the audio caption generated by Qwen2-Audio. Notably, these samples of audio carrying speech information are split into the AVU-Specific subset in our dataset generation pipeline.

### D.6 SPEECH COMPREHENSION CAPABILITY

We have conducted an exploration into the speech capabilities of Dolphin. Firstly, we discovered that Dolphin has a certain level of speech recognition ability. From the perspective of performance, the speech capabilities are not bad. In Table 3, Dolphin outperforms other audio-centric models by a large margin. We attribute the result to the fact that the generated AVU dataset contains a significant amount of speech data.

Specifically, when we use Qwen2-Audio to generate captions, it will perform real-time transcription for speech. We specifically calculated that samples containing speech transcription, the results account for 39.6% of all audio captions. These speech-related data help enhance speech abilities. As shown in Figure. 8 They contain a lot of speech information in the captions and are generally categorized into the AVU-Specific subset.

Similar findings that speech-centric training datasets could help enhance the speech recognition of audio encoders have been concluded by some prior works. For example, in LTU (Gong et al., 2023b), LTU-Speech and LTU-audio utilize freeze audio encoders (AST encoder using CAV-MAE (Gong et al., 2022) objectives) trained with audio and speech data, respectively. Considering the performance, LTU-Speech outperforms LTU-Audio by a large margin on Speech Question datasets (93% >69%).

## E DATASET DETAILS

### E.1 THE MOTIVATION OF THE AVU-DATASET

We list the purpose of the proposed AVU-dataset as follows: (1) There is a lack of relevant datasets. Currently, there are no large-scale audio-visual captioning and instruction tuning datasets available. Existing methods have only been trained on vision-language and audio-language datasets, resulting in a deficiency of audio-visual alignment, which consequently leads to suboptimal performance in audio-visual tasks. (2) We analyze that one of the primary reasons existing models overlook the audio modality is that, in most cases, audio does not provide more information than video. Therefore, one of our significant innovations lies in our dataset, which specifically selects samples where audio conveys more information than video. We have transformed this audio-specific information into question-and-answer pairs, effectively addressing the issue of AV-LLM's neglect of audio. (3) Existing audio-visual

datasets exhibit diverse annotation formats (bounding boxes, timestamps, masks). Consequently, we have standardized the input and output formats for audiovisual tasks such as AVE, AVL, AVQA, and AVVP, facilitating the training of AV-LLM. Additionally, we have provided a dataset with fine-grained temporal and spatial granularity (AVU-tasks).

### E.2   THE CONTRIBUTION OF THE AVU-DATASET

The contribution of our AVU-dataset is summarized as follows: (1) The AVU dataset addresses the current lack of large-scale, high-quality audiovisual instruction tuning datasets within the community. (2) The AVU dataset features a rich variety of subsets, effectively addressing the issue of AV-LLM neglecting the audio modality and significantly reducing the occurrence of hallucinations. (3) The introduction of meta-information in the AVU dataset, which is categorized based on AV consistency, allows for the extraction of modality-specific information (AVU-specific) and the generation of negative samples (AVU-negative). This approach can be widely applied in the process of creating other datasets.

### E.3   PROMPT TEMPLATES.

The prompt templates are shown in Figure 9, Figure 10 and Figure 11.

**[Audio-video information]:**
This is the **detailed caption of the video**: {...}; and **the meta-information**: {...}. This is the **detailed caption of the audio:** {...}; and **the meta-information:** {...}. These above contain important information that needs to be read carefully.*

**I . AVU-Pretrain**

**INPUT:**

**[Audio-video information]**

Your task is to generate a detailed caption for an audio-video segment based on provided meta-information and detailed descriptions. Ensure the caption is comprehensive and accurate, using only the provided information without adding any fictional content.

- - - - - - - - - - - - - - - - - - - - - - - - - - - - - - - - - - - - - - - - - - - - - - - - - -

**EXAMPLE:**

➢ The video starts with a woman standing in the kitchen, holding a knife and a potato. She begins slicing the potato thinly, showcasing her skill and precision. The camera captures her movements as she continues slicing, demonstrating the food preparation process. Soft ambient sounds fill the kitchen, including faint simmering from a nearby stove and occasional utensil clinks.

**II. AVU-MultiQA**

**INPUT:**

**[Audio-video information]**

Your task is to generate ten pairs of instructions and responses based on the provided meta-information and detailed captions of the audio and video content. Each pair should simulate a dialogue that focuses on various aspects of the content without introducing information not explicitly mentioned. The purpose of this exercise is to fine-tune a language model so that it can generate accurate and relevant responses.

The QA pair should start with **"Q:"** and contain an instruction related to the audio and video, while the second line should start with **"A:"** and provide a response to the instruction.

Please ensure that your instructions are diverse and of high quality. Avoid making assumptions or adding details that are not explicitly given.

- - - - - - - - - - - - - - - - - - - - - - - - - - - - - - - - - - - - - - - - - - - - - - - - - -

**EXAMPLE:**

➢ **Q:** Describe how the video integrates visual steps and audio tips to guide viewers in making pancakes.
  **A:** The video combines close-up visuals of pancake preparation with descriptive audio commentary, emphasizing quick cooking and stacking instructions.

➢ **Q:** Provide a summary of the video's approach to combining visual demonstrations and audio explanations.
  **A:** The video seamlessly integrates detailed visual demonstrations of pancake preparation with insightful audio explanations, offering viewers a comprehensive guide to achieving perfect pancakes.

Figure 9: Prompt templates for generation of AVU-dataset subsets.

**[Audio-video information]:**
This is the **detailed caption of the video**: {...}; and **the meta-information**: {...}. This is the **detailed caption of the audio:** {...}; and **the meta-information:** {...}. These above contain important information that needs to be read carefully.*

**II. AVU-Specific**

**INPUT:**

**[Audio-video information]**

You have to finish the following two tasks. First, carefully compare the information typically included for audio versus video. Second, Identify information that is included for audio but not for video, and generate a QA pair for each. Similarly, identify information that is included for video but not for audio, and generate a QA pair for each.

Each pair should simulate a dialogue that focuses on various aspects of the content without introducing information not explicitly mentioned. The purpose of this exercise is to fine-tune a language model so that it can generate accurate and relevant responses.

The QA pair should start with **"Q:"** and contain an instruction related to the audio and video, while the second line should start with **"A:"** and provide a response to the instruction.

Please ensure that your instructions are diverse and of high quality. Avoid making assumptions or adding details that are not explicitly given.

- - - - - - - - - - - - - - - - - - - - - - - - - - - - - - - - - - - - - - - -

**EXAMPLE:**

➢ **Q:** What kitchen tools or utensils are prominently featured in the **video** during pancake preparation?
 **A:** The video showcases basic kitchen tools such as a mixing bowl, whisk, spatula, and frying pan, demonstrating their use in making pancakes.

➢ **Q:** Discuss the overall tone and style of the English **audio** commentary. How does it enhance the tutorial?
 **A:** The commentary's friendly and instructive tone makes pancake-making feel approachable and enjoyable, encouraging viewers to try the recipe themselves.

**IV. AVU-Negatives**

**INPUT:**

**[Audio-video information]**

The information provided for audio and video does not fully match. Find out what the audio and video information does not match. Create negative question-answer pairs where the question asks for information not included in the provided data. Provide negative responses or rejections indicating the information is unavailable. Don't design questions that can be answered with yes or no.

The QA pair should start with **"Q:"** and contain an instruction related to the audio and video, while the second line should start with **"A:"** and provide a response to the instruction.

- - - - - - - - - - - - - - - - - - - - - - - - - - - - - - - - - - - - - - - -

**EXAMPLE:**

➢ **Q:** Can you describe the sound of the frog in the audio?
 **A:** The color of the frog is not mentioned in the audio. The background audio is funny music.

Figure 10: Prompt templates for generation of AVU-dataset subsets.

**[Video caption information]:**
This is the **detailed caption of the video**: {...}.These above contain important information that needs to be read carefully.*

**Meta-Information**

**INPUT:**

**[Video caption information]**

You are an AI assistant for video information integration. I will provide you with a caption about a video, and please extract some meta information from it, including "subject"(the main character in the picture), "scene"(the scene and the events in the video), "location" (the exact location of the video. Not geographical location), "action" (the movements of the main character in the video), and "emotion" (the emotion of the subject or the video). If the information is not available or not explicitly mentioned, please output the words "None". Avoid making assumptions or adding details that are not explicitly given.

- - - - - - - - - - - - - - - - - - - - - - - - - - - - - - - - - - - - - - - - - - -

**EXAMPLE:**

➢ **Original video caption:** The video shows a large crowd of people gathered in a stadium, with some individuals holding flags and cheering. The flags are positioned in the foreground, while the crowd is in the background. The people in the crowd are standing and cheering, with some individuals holding up their hands. The stadium is filled with people, and the atmosphere is lively and energetic.
**Meta-information:**
Subject: Large crowd of people, individuals with flags
Scene: Stadium
Location: Stadium with a large crowd
Action: Individuals holding flags and cheering, crowd standing and cheering, some individuals holding up their hands
Emotion: Lively and energetic atmosphere

**[Audio caption information]:**
This is the **detailed caption of the audio**: {...}.These above contain important information that needs to be read carefully.*

**Meta-Information**

**INPUT:**

**[Video caption information]**

You are an AI assistant for audio information integration. I will provide you with a caption about an audio, and please extract some meta information from it, including "subject"(the main character in the audio), "scene"(the scene and the events in the audio), "location" (the exact location of the audio. Not geographical location), "action" (the movements of the main character in the audio), and "emotion" (the emotion of the subject or the audio). If the information is not available or not explicitly mentioned, please output the words "None". Avoid making assumptions or adding details that are not explicitly given.

- - - - - - - - - - - - - - - - - - - - - - - - - - - - - - - - - - - - - - - - - - -

**EXAMPLE:**

➢ **Original video caption:** A woman is speaking in a clear and confident tone. She is describing a car dealership and the various cars that are available. She mentions the different features of each car and the benefits of purchasing from the dealership. The music is playing in the background.
**Meta-information:**
Subject: A woman
Scene: A car dealership description
Location: None
Action: Speaking in a clear and confident tone, describing cars and their features
Emotion: Confident
Background: Music playing

Figure 11: Prompt templates for generation of meta-information.

