# OpenReview forum: "Aligned Better, Listen Better for Audio-Visual Large Language Models"
_ICLR.cc/2025/Conference — ICLR 2025 Poster_

### Official Review · Reviewer_f89q · 2024-10-27

**Soundness:** 2
**Presentation:** 2
**Contribution:** 2
**Rating:** 5
**Confidence:** 5

**Summary:**

This paper investigates the capabilities of audio-visual large language models (AV-LLMs) to enhance their reasoning and understanding capabilities. As existing AV-LLMs tend to neglect audio information, this paper addresses the issue from two perspectives: model architecture and dataset. For model architecture, the authors enhance both spatial and temporal alignment of AV-LLMs by proposing an audio-visual multi-scale adapter for aggregating multi-scale information and proposing audio-visual interleaved merging, respectively. For the dataset, this paper proposes a large-scale caption & instructional dataset using existing audio-visual data sources. Experimental results show that the proposed model achieves favorable performance in both audio-visual understanding and audio understanding tasks.

**Strengths:**

- The motivation of this work, which identifies the weaknesses of AV-LLMs and aims to solve them from two different perspectives, is a sound approach to advancing research in this area.
- The approach to enhancing spatial and temporal alignment in audio-visual LLMs is innovative.
- Constructing an audio-visual caption and instructional dataset is beneficial for researchers, as there is a lack of such datasets.

**Weaknesses:**

**The main text requires further refinement. It contains typos, broken sentences, and inconsistent tenses. The reviewer has identified only some of these issues:**
 - L67: VideoLlama is mentioned twice.
 - L158: "clap" should be checked for correctness.
 - L231: "detailed" is misspelled as "detrailed."
 - L414 contains a broken sentence.
 - The right figure in Figure 1 is not explained in the main text.
 - There is a typo in the right figure of Figure 2, "his audio."
 - The text should use "/citep" for citations.

**The reviewer is concerned about the reliability of the dataset. Since the paper proposes a large-scale dataset, it should include a more detailed explanation, such as dataset statistics. The reviewer points out some missing or problematic aspects that lessen the dataset's reliability:**
- The prompt templates for constructing the meta information are not provided. These prompts are crucial as they differentiate dataset types and help manage noise in this automatically generated dataset.
- In Figure 6, AVU-specific, although the questions differ, the answers are identical.
- In Figure 9, the question asks about the sound of a frog, yet the answer discusses an unrelated aspect of color, highlighting the dataset's noisiness.
- To address concerns about the dataset's reliability and its claim as a benchmark, human verification of the dataset is necessary. If the dataset is noisy, researchers might hesitate to use it for evaluating models.

**The comparison experiments are not thoroughly conducted. Since the paper focuses on improving the audio-visual understanding of AV-LLMs, it should include comparisons with existing high-performing AV-LLMs. Here are several models that the paper should have considered:**
- FAVOR: https://arxiv.org/pdf/2310.05863
- video-Salmon: https://arxiv.org/pdf/2406.15704
- PandaGPT:https://arxiv.org/abs/2305.16355
- OneLLM: https://arxiv.org/pdf/2312.03700

**The reliability of the model's design and training is questionable. The inconsistencies and errors in the paper amplify these concerns:**
- The notations in Figure 2 and the main text differ, making it hard to understand the model's mechanism.
- What does the superscript “i” stand for in all notations? And what is the difference from the superscript “1” in L178?
- In Figure 1, how does the Dolphin model recognize the words a man says using the ImageBind audio encoder? Doesn't the ImageBind audio encoder take environmental sound as an input, not speech?
- In L430, the authors mention that AST was used, but do not explain how they trained or integrated this model.
- Table 6 not explained in the main text.

**Questions:**

- L32: Does the audio modality prove crucial for comprehensive understanding? Could you substantiate this claim?
- Does reason (3) on L72 contradict the starting paragraph of the introduction, where the authors assert that audio is crucial for video understanding? Could the authors provide examples of when audio is crucial versus when it may be less informative than visual data?
- In Table 2 and Table 3, did Dolphin use unimodal signal as an input, or use both of multimodal signal for unimodal task?
- L67: It appears that the model trained with audio converted to text performs favorably. How would the model perform with video + audio (converted to text)? Could this combination outperform the Dolphin model? Could the authors conduct this experiment?

---

### Official Review · Reviewer_gDzU · 2024-11-03

**Soundness:** 3
**Presentation:** 3
**Contribution:** 2
**Rating:** 6
**Confidence:** 4

**Summary:**

This paper discusses the importance of audio-visual large language models (AV-LLMs) in multimodal video understanding, with a particular emphasis on the use of audio information. The paper proposes a fine-grained AV-LLM model called Dolphin, which ensures comprehensive and accurate video understanding by aligning audio and video in both spatial and temporal dimensions. To better define the task, this work proposed a related dataset(AVU) and benchmark(AVU-Bench), that contains 5.2 million diverse data pairs (video, audio, questions, answers), and a novel data partitioning strategy is introduced. Experimental results show that Dolphin performs well in audio-visual understanding and effectively reduce hallucinations.

**Strengths:**

The method is soundness. The author put forward a fine grand alignment method, adding visual tokens audio, and special temporal Temporal tokens to achieve better alignment.
The.

This paper put forward a comprehensive dataset with a promising data processing pipeline and obtained large-scale data.

The paper gives a benchmark based on the task definition and its dataset and compares the baseline methods.

Extensive experiments demonstrate that Dolphin significantly improves audio-visual comprehension and is effective in reducing errors related to audio neglect.

**Weaknesses:**

1. The experiment is comprehensive but the baseline is weak. The method mentioned VideoLLAMA2, but the experiment seems only to compare the result with VideoLlaMA1. Adding more comparisons against these baselines would be more persuasive.

2. The author mentioned that AVU could reduce the hallucination; while the related analysis is not included in the experiments.

3. The meaning of “fine-grained spatial modeling” lack of definition. Please provide a clear definition or explanation of "fine-grained spatial modeling" in the context of their work.

4. Although the author compares video and audio captions separately, more experiments on other audio-visual datasets are expected.
Many any-to-any models can have a visual-audio understanding ability. What is their performance on the given tasks?

**Questions:**

Please refer to the weakness.
Overall, I think this article is quite comprehensive, but in this era of a large number of LLM works, I think this work needs to be supplemented with more comparisons to prove that this work is novel enough to be published in ICLR.

---

> ### Comment · Reviewer_gDzU · 2024-11-26
> **Feed back to author**
>
> Dear author,
>
> Here I gave a quick feedback first, since the review period extended, I expect more discussion w/o more experiments needed.
> I appreciate the author's attitude toward adding the experiments. In general, I personally expect more explanations and descriptions of the given experiments.
> However, I already raised the score to 5->6 since most of my concerns are well addressed. While the data scale and potential use still be a limitation from my perspective.
>
> Best

---

### Official Review · Reviewer_JTeq · 2024-11-03

**Soundness:** 3
**Presentation:** 2
**Contribution:** 3
**Rating:** 6
**Confidence:** 5

**Summary:**

The authors propose an audio-visual LLM Dolphin, which consists of a multi-scale adapter for spatial alignment and an interleaved merging module for temporal alignment. A large-scale audio-visual caption&instruction-tuning dataset AVU is also proposed, including 5.2M video-audio-qa tuples. Training on the proposed dataset, the proposed method achieves state-of-the-art performance on several audio-visual, audio, and video benchmarks compared with existing audio-visual LLMs.

**Strengths:**

1. The curation process of the AVU dataset looks sound and reasonable. The authors integrate several open-source and commercial LLMs into the data pipeline to generate high-quality audio-visual captions and divide the dataset into several parts based on audio-visual consistency. The community now is facing a shortage of a large-scale audio-visual instruction-tuning dataset. The proposed dataset, along with the data curation procedure, will help the following research in the related field.

2. The results show the proposed method outperforms several previous audio-visual LLM on audio, video, and audio-visual benchmarks.  Apart from caption and question-answering, it also excels in some closed and open-ended audio tasks, which makes the framework more applicable.

3. The ablations are comprehensive. Each component is well-ablated and clearly verified. The authors also conduct numerical analysis on the impact of the proposed dataset.

**Weaknesses:**

1. The method is trivial and questionable. The entire framework consists of three parts: audio and visual encoders with injected multi-scale uni-modal and multi-modal adapters, a cross-modal attention block to perform temporal integration, and a Vicuna as the decoder. The audio-visual adapters and the cross-modal attention have been proposed and utilized in many previous works[1-3], and the pipeline of training an audio-visual LLM is also not novel. The data pipeline for generating audio-visual captions is also been utilized by several previous methods[4-5]. Besides, the description of the model architecture is vague, many details are missing and the rationale of some model designs is unclear. Please see the question part below in detail.

2.  Speech is neglected in the model architecture designs. Since the audio feature is semantic and high-level, while the speech feature is low-level and dense, it is a common way to model the audio and speech separately via different encoders, such as [4, 6]. Besides, how does the proposed model outperform baseline methods on the speech recognition task as shown in Table 3 when no speech encoder or dense feature is involved? What does the model perform when compared with some speech-centric models?

3. The application scenarios are limited. It seems that the proposed method is only suitable for audio-visual correspondence videos since the training dataset is constructed by at least medium-level AV consistency videos, while the low-level AV consistency data is used for negative samples, yet 1). how to decide whether an in-the-wild video is suitable for the model to infer? and 2). what is the purpose of aligning audio and visual encoders using high AV consistency videos? I believe the alignment stage is more likely to align the audio and visual encoder with the text decoder rather than align the audio encoder with the visual encoder. What will happen if videos with low AV consistency are introduced for training?

4. Audio-visual capabilities are not fully probed. Some audio-visual tasks are not tested, such as audio-visual caption, audio-visual speech recognition, and audio-visual sound source detection as the previous method [6] does. I suggest the authors conduct experiments on these benchmarks and compare the proposed method with [6] to show the model's capability more comprehensively.

Reference:

[1] Lin, Yan-Bo, et al. "Vision transformers are parameter-efficient audio-visual learners." Proceedings of the IEEE/CVF Conference on Computer Vision and Pattern Recognition. 2023.
[2] Tian, Yapeng, Dingzeyu Li, and Chenliang Xu. "Unified multisensory perception: Weakly-supervised audio-visual video parsing." Computer Vision–ECCV 2020: 16th European Conference, Glasgow, UK, August 23–28, 2020, Proceedings, Part III 16. Springer International Publishing, 2020.
[3] Li, Guangyao, et al. "Learning to answer questions in dynamic audio-visual scenarios." Proceedings of the IEEE/CVF Conference on Computer Vision and Pattern Recognition. 2022.
[4] Chen, Sihan, et al. "Vast: A vision-audio-subtitle-text omni-modality foundation model and dataset." Advances in Neural Information Processing Systems 36 (2023): 72842-72866.
[5] Wang, Yi, et al. "Internvideo2: Scaling video foundation models for multimodal video understanding." arXiv preprint arXiv:2403.15377 (2024).
[6] Sun, Guangzhi, et al. "video-SALMONN: Speech-enhanced audio-visual large language models." arXiv preprint arXiv:2406.15704 (2024).

**Questions:**

1. Considering the selected audio and visual encoders are far smaller than the LLM (ViT-L and AST), why not directly train these encoders to achieve better performance since the 7b/13b LLM is also involved in training in the instruction-tuning stage?

2. Why select ViT-L, AST, and Vicuna as encoders and decoders when tons of more powerful alternatives are available (such as SigLIP, InternViT for image, Beats, Whisper encoder for audio, and Qwen, llama3, mistral for LLM)? Is there any ablation?

3. Why not use some video encoders to perform visual encoding both for the Dolphin model and the data curation pipeline? Is there any ablation?

4. For the temporal integration, how does the proposed bi-directional cross-attention block 'enhance the audio-visual information exploitation of AV-LLM' as the author claims? What I see is just an attention block to perform cross-modal interaction for global features, yet how to model the temporal relationships, is positional encoding or RoPE being used? How to inject the so-called 'temporal integration information' into the dual-path framework? The descriptions are too vague and need to be improved.

5. What is the connector between the audio/visual encoder and LLM decoder? Q-former or linear projection? Is there any ablation?

6. How does the model tackle uni-modal tasks since the fine-grained alignment seems to be mandatory? For videos that missing the auditory part, will a modality mask perform on the input of the LLM decoder and the cross-modality integration module (both spatial and temporal)? For videos with semantic-irrelevant auditory parts, how does the model resist the potential negative information brought by the auditory modality?

7. For the experiments, the authors only compare the proposed method with audio-visual LLMs, how much is the performance gap between the proposed AV-LLM with some uni-modal models?

---

### Official Review · Reviewer_pdCN · 2024-11-03

**Soundness:** 3
**Presentation:** 3
**Contribution:** 3
**Rating:** 8
**Confidence:** 4

**Summary:**

This paper introduces a new audio-visual LLM model called Dolphin and a new audio-visual dataset AVU. The authors discuss the existing problem with video LLMs, which is, how they often ignore the audio information present in the video and only attend to the visual information while understanding videos. The authors claim that the models do not learn any alignment between the audio and visual information in the video, which is the reason for this behavior of video LLMs. Hence the authors design the Dolphin model, which aligns the audio and visual information both spatially and temporally before feeding them to the LLM. Specifically, they use multi-scale vision transformers to extract visual features at different scales and apply cross-attention with audio features at each scale. These
features are again merged with the global visual representation using another cross-attention. Then temporal cross-attention is applied between these features bi-directionally to obtain visual-contextualized audio tokens and audio-contextualized visual tokens. This is fed to the LLM for the downstream task.

Since most existing video datasets focus mainly on visual content, the authors have introduced a new audio-visual dataset by using existing unimodal datasets and leveraging LLMs to generate modality-specific question-answer pairs. They generate different types of questions and answers based on metadata correspondence of the audio and visual inputs by prompting LLMs. The experiments are designed to test the new model architecture on existing video QA datasets and other unimodal tasks such as captioning and classification.

**Strengths:**

1. The problem addressed by the authors is an important one. Most video-related datasets and models indeed ignore the information present in the audio almost completely. Hence this work is an important one to fill this research gap.

2. The proposed model architecture achieves better results on existing video QA datasets and the ablation studies show the importance of spatial and temporal alignment layers introduced in the architecture.

3. The dataset is large-scale and can be significant to the community to advance audio-visual understanding.

4. The usefulness of the dataset is shown by comparing video llama trained with and without the AVU dataset.

**Weaknesses:**

1. The entire pipeline in the dataset generation is LLM-based. There are no discussions about the efficiency of the pipeline, hallucination effects, or error propagation in the dataset creation process.

2. The authors claim in a lot of places in the paper that there is a significant reduction in hallucinations using their model and dataset. They design an AVU-negatives subset to train the model to say no to some questions.  However, the experiments are not designed to validate this claim in any manner. While Dolphin may outperform certain models, it is unclear whether the hallucination is reduced as there are no metrics or definitions to evaluate this. It is a tall claim without any experimental results to say that hallucinations are reduced.

3. Minor comment: Clotho-V2 which was is used as a dataset for training is not referenced.

**Questions:**

1.  What are the effects of using pre-trained models to create a pipeline for various captioning and QA creation steps? What if any of the models hallucinated? Was there some kind of quality check done?

2. I am intrigued by some of the examples of the dataset that has absolute time information such as "What time does the train whistle blow?" and the model providing an answer. Do these models understand the concept of time and seconds?

---

### Meta-Review · Area_Chair_hDuV · 2024-12-19

**Metareview:**

The paper presents Dolphin, an audio-visual (AV) reasoning framework, that aligns audio and visual modalities in a fine-grained manner using transformer cross-attention at multiple spatial scales for each audio-visual frame. The proposed method uses a large language model to take the aligned AV features as tokens to produce reasoning responses. The paper further proposes a dataset: Audio-Visual Understanding (AVU), by putting together multiple AV benchmarks, and providing additional annotations using unimodal foundation models; the annotations are improved using expert models, LLMs based on consistency, and manual verification. Experiments are provided on audio-visual datasets over QA tasks and demonstrate promise against recent prior methods.

The paper received four mixed reviews, mainly inclined favorably. All the reviewers agree on the importance AV alignment in modern multimodal LLMs and support the AVU dataset the paper proposes. However, there are also concerns regarding the reliability (pdCN, gDzU, f89q) of annotations in the AVU dataset, especially given it is generated automatically using expert models. Reviewers also pointed out issues with regards to the technical contribution for multimodal alignment that has similarities to many prior works (JTeq, f89q), lack of experiments into various aspects of the model, dataset, and capabilities (JTeq, gDzU, f89q).

**Additional Comments On Reviewer Discussion:**

The paper was discussed extensively between the reviewers and the authors, with the authors providing detailed point-by-point explanation of the concerns, as well as providing new experimental comparisons and results. Some of the key discussion points are summarized here.

1. Hallucination in the expert models during dataset generation (pdCN, gDzU, f89q): To address this concern, authors provided performance of their model on video and audio hallucination benchmarks, where the results show minor improvements in avoiding video hallucination (against Video LLaMA2) and a significant improvement in mitigating audio hallucination.

2. The novelty of the proposed architecture being straightforward or very similar to prior works (JTeq, f89q): The authors provide clarifications on the contributions. AC also observes the ablation studies provided in Table 5 speak out the importance of each component in the model. However, AC also thinks that better insights could have been provided to support the various design choices made in the architecture.

3. Missing experiments, comparisons to state-of-the-art models, dataset annotation details, and model capabilities (JTeq, gDzU, f89q): Authors have provided many new results during the discussions, including results to VideoLLaMA2, Avicuna, Video-Salmon, PandaGPT, etc., showing improvements. One facet where the model struggles is perhaps speech recognition (as shown in the Table in response to Q2 of Reviewer JTeq).

Overall, AC thinks the paper makes a good contribution from a dataset perspective that may be useful for training future MLLMs. While the technical contribution is weak, it appears to have some novel components that are empirically shown to lead to strong performance. Thus, AC recommends accept.

---

### Decision · Program_Chairs · 2025-01-22

Accept (Poster)